# Early Cholecystectomy after Endoscopic Retrograde Cholangiopancreatography Is Feasible and Safe

**DOI:** 10.3390/healthcare12141407

**Published:** 2024-07-15

**Authors:** Çağrı Büyükkasap, Deniz Algan, Nigar Balakji, Onur Metindoğan, Ali Karataş, Aydın Yavuz, Hüseyin Göbüt, Kürşat Dikmen, Murat Kekilli, Hasan Bostancı

**Affiliations:** 1Department of General Surgery, Gazi University Faculty of Medicine, 06500 Ankara, Turkey; onur.metindogan@gazi.edu.tr (O.M.); aydinyavuz@gazi.edu.tr (A.Y.); hgobut@gazi.edu.tr (H.G.); kursatdikmen@gazi.edu.tr (K.D.); hasanbostanci@gazi.edu.tr (H.B.); 2Gazi University Faculty of Medicine, 06500 Ankara, Turkey; zeynepdeniz.algan@gazi.edu.tr (D.A.); nigar.balakci@gazi.edu.tr (N.B.); 3Department of Gastroenterology, Gazi University Faculty of Medicine, 06500 Ankara, Turkey; alikaratas@gazi.edu.tr (A.K.); muratkekilli@gazi.edu.tr (M.K.)

**Keywords:** choledocholithiasis, ERCP, laparoscopic cholecystectomy, early cholecystectomy, perioperative outcomes

## Abstract

Laparoscopic cholecystectomy (LC) following endoscopic retrograde cholangiopancreatography (ERCP) is the preferred treatment for cholelithiasis with common bile duct stones. However, the optimal timing of LC after ERCP remains controversial. This study aimed to identify the ideal time interval between ERCP and LC. Data from patients who underwent LC after ERCP between November 2016 and August 2022 were retrieved from the electronic medical information system. The patients were categorized into early cholecystectomy (within 72 h of ERCP) and delayed cholecystectomy (after 72 h). The impact of the time interval between ERCP and LC on perioperative outcomes was analyzed. A total of 197 patients were included in the study, with 45 undergoing early cholecystectomy and 152 undergoing delayed cholecystectomy. No significant differences in the preoperative characteristics or operative morbidity were observed between the groups (*p* = 0.286). However, a significant correlation was found between the time interval from ERCP to LC and the total length of stay (r = −350, *p* < 0.001). The findings suggest that early cholecystectomy after ERCP is feasible and safe, and performing LC within the first 72 h does not adversely affect postoperative outcomes.

## 1. Introduction

Cholelithiasis is one of the most common surgical pathologies, and laparoscopic cholecystectomy (LC) has long been recognized as the standard treatment for symptomatic cholelithiasis. In 10–20% of patients with cholelithiasis, concomitant choledocholithiasis is present [1]. Accepted approaches for the treatment of choledocholithiasis include laparoscopic common bile duct exploration (LCBDE), the laparoendoscopic rendezvous method (LERV), and LC after endoscopic retrograde cholangiopancreatography (ERCP). Both LCBDE and LERV allow for the simultaneous treatment of cholelithiasis and choledocholithiasis, thereby preventing recurrent biliary events. The guideline for the treatment of common bile duct stones issued by the European Association for Endoscopic Surgery recommends both LERV and LCBDE for the treatment of choledocholithiasis in centers where these techniques are applicable [2]. However, many centers cannot apply these methods due to the difficulties in their application and the need for special training and experience [3,4,5]. Currently, preoperative ERCP followed by laparoscopic cholecystectomy is the most commonly used method in the treatment of choledocholithiasis.

The timing of LC after ERCP in patients with choledocholithiasis remains a matter of debate. Numerous studies on the time interval between ERCP and LC recommend early LC [6,7,8]. These studies have shown that early LC can reduce perioperative complications, the length of stay, and hospital costs. However, the concept of “early” used in the literature varies between institutions. Recurrent biliary events (RBEs) are one of the significant problems that occur after ERCP treatment of choledocholithiasis, leading to increased operative morbidity in this patient group [9,10]. In a study of 529 patients who underwent preoperative ERCP and LC for choledocholithiasis, Bergeron et al. [10] showed that 28.5% of the patients developed RBEs during the waiting period for the operation; 13% developed choledocholithiasis, which occurred at a median postoperative day of 42 (range of 25–94 days). In many centers, early cholecystectomy is preferred within the first 3 days before the onset of inflammatory changes caused by ERCP [11,12]. The inflammation caused by ERCP and subsequent bacterobilia may make Callot dissection difficult, which is the primary rationale for early cholecystectomy. In the study by Aziret et al. [12], less bacterobilia was found in the early cholecystectomy group compared to the delayed cholecystectomy group, and polymorphonuclear lymphocyte/mononuclear lymphocyte infiltration and fibrosis/collagen deposits also occurred less in this group according to histopathologic examination. Performing ERCP and cholecystectomy on the same day is not preferred in many centers due to concerns that insufflation during endoscopy may cause visual impairment during the operation and the need to monitor the patient for possible ERCP complications. However, studies suggest that cholecystectomy can be safely performed immediately after ERCP [4,13]. Despite the literature emphasizing early cholecystectomy, many studies have shown that delayed cholecystectomy is performed more frequently than early cholecystectomy [8,14,15].

This study aimed to determine the ideal time interval between ERCP and LC by evaluating the perioperative results of our patients who underwent ERCP and subsequent cholecystectomy for choledocholithiasis.

## 2. Materials and Methods

### 2.1. Study Population

This study was approved by our local Ethics Committee (Approval Number: 03.04.2023-287). Patients who underwent ERCP and subsequent cholecystectomy in our hospital between November 2016 and August 2022 were identified by searching the electronic medical information system. Patients younger than 16 years of age, those without a plan for cholecystectomy after ERCP (gallbladder in situ), those who had ERCP performed postoperatively, those with diagnoses other than choledocholithiasis suspected during ERCP (such as malignancy or chronic pancreatitis), those who underwent emergency surgery due to complications during ERCP, and those with a previous Roux-en-Y anastomosis were excluded from the study. The time between ERCP and the operation was calculated according to the last ERCP performed before the operation. The patients who underwent cholecystectomy within 72 h after ERCP were classified as the early cholecystectomy group, and those operated on after 72 h were classified as the delayed cholecystectomy group. Statistical analysis was also performed separately for the patients who underwent cholecystectomy within the first 24, 48, and 72 h (Figure 1).

### 2.2. Procedures

The diagnoses of acute cholecystitis and acute cholangitis were defined in accordance with the 2018 Tokyo guidelines [16,17]. The diagnosis of acute pancreatitis was defined as an increase in the serum lipase level of more than three times the upper limit with clinical suspicion. The diagnosis of patients with clinically suspected choledocholithiasis was routinely clarified using MRCP unless contraindicated. All ERCP procedures were performed in the semi-lateral position and under deep sedation. The common bile duct was cannulated, and a standard length sphincterotomy was performed. After sphincterotomy, stone removal was performed with a balloon catheter. A biliary stent was placed in patients with purulent discharge and suspected residual stones. A pancreatic stent was applied in patients whose main pancreatic duct was cannulated. Patients with biliary stenosis underwent dilatation with a 10 or 12 mm balloon. Laparoscopic cholecystectomy was performed with the standard four-trocar technique. In case of conversion to open surgery, a right subcostal incision was made. A drain was placed in the subhepatic area in case of suspected bile leakage and bleeding risk.

### 2.3. Data Collection

The data collected included age, gender, American Society of Anesthesiologists (ASA) scores, preoperative laboratory values, preoperative history of cholecystitis, cholangitis, pancreatitis, number of hospitalizations for biliary causes, the diameter of the common bile duct on magnetic resonance imaging before ERCP, number of preoperative ERCPs, ERCP findings, preoperative biliary stent and balloon dilatation, intraoperative additional biliary procedures, postoperative complications of grade 3 and above according to the Dindo–Clavien system, total and postoperative length of stay, need for reoperation, and need for ERCP in the postoperative period.

### 2.4. Statistical Analysis

The data were analyzed using IBM SPSS V23. The conformity of the data to normal distribution was assessed using the Kolmogorov–Smirnov test. The Pearson chi-square test was used to analyze categorical data. The Mann–Whitney U test was used to analyze nonparametric data. ROC analysis was performed to find cut-off values to differentiate complications. Spearman’s rho correlation coefficient was used to analyze the relationship between continuous parameters that did not fit the normal distribution. Logistic regression analysis was used to identify the risk factors affecting complications. The analysis results are presented as the frequency (percentage) for categorical variables, mean ± standard deviation, and median (minimum–maximum) for quantitative variables.

## 3. Results

### 3.1. Study Population and Baseline Characteristics

Between November 2016 and August 2022, 220 patients who underwent ERCP and subsequent laparoscopic or open cholecystectomy were evaluated. Seventeen patients were excluded, because ERCP was performed postoperatively. In four patients, the common bile duct could not be cannulated during ERCP, necessitating intraoperative common bile duct exploration. Additionally, two patients were diagnosed with pancreatic malignancy on imaging studies. Thus, 197 patients met the inclusion criteria and were included in the analysis. The preoperative characteristics of these patients are shown in Table 1.

### 3.2. Early vs. Delayed Cholecystectomy

The characteristics of patients who underwent early and delayed cholecystectomy are shown in Table 2. There were no significant differences in age (59.29 vs. 55.62, *p* = 0.255). A total of 57.78% of early cholecystectomy and 40.13% of delayed cholecystectomy patients are female (*p* = 0.802). The ASA scores were also similar between the two groups (24, 21 vs. 95, 57, *p* = 0.269).

### 3.3. Perioperative Outcomes

Pancreatitis was detected in 46 patients and cholangitis in 41 patients. Choledocholithiasis was identified with MRCP in 122 patients without cholangitis or pancreatitis. There was no history of cholecystitis, cholangitis, or pancreatitis in 74 of the studied patients. While 151 patients underwent one session of ERCP preoperatively, 197 patients underwent ERCP a total of 261 times. The most common reason for repeat ERCP was the presence of residual stones or failed cannulation. Sphincterotomy was performed in all but two patients due to unsuccessful common bile duct cannulation. Balloon dilatation was performed in one patient with biliary stenosis detected with ERCP. Forty-two patients underwent biliary stenting due to purulent discharge from the common bile duct or residual calculi. Plastic stents were used for biliary stenting. The ERCPs of three patients showed no choledocholithiasis.

### 3.4. Surgical Details and Complications

Among the study participants, three patients underwent open surgery due to previous upper abdominal surgery and two due to failure to remove stones via ERCP. Laparoscopic cholecystectomy was initiated in 192 patients, with 16 requiring conversion to open surgery due to adhesions or to control bile leakage and bleeding. Early cholecystectomy was performed in 45 patients, with 80% of these operations occurring within the first 24 h. In the delayed cholecystectomy group, the mean interval between ERCP and surgery was 68.61 days. Only 23 patients were operated on between day 14 and day 30, while the waiting period was more than 6 weeks in 98 patients. Perioperative complications and operation-related morbidity were similar between the early and delayed groups (2.2% vs. 4.6%, *p* = 0.487 and 8.9% vs. 12.5%, *p* = 0.51). When the effect of time between ERCP and cholecystectomy on perioperative outcomes was analyzed using logistic regression analysis, no significant difference was found in terms of operation-related morbidity (Table 3). The postoperative length of stay increased with an increase in the waiting time after ERCP, but the relationship was not significant (*p* = 0.067). However, there was a statistically significant positive correlation between the waiting time and total length of stay (Table 4).

### 3.5. Length of stay and Recurrent Biliary Events

The postoperative length of stay increased with the waiting time after ERCP, but this was not statistically significant (*p* = 0.067). However, there was a significant positive correlation between the waiting time and total length of stay. Intraoperative biliary procedures were performed in six patients. Three patients underwent open common bile duct exploration due to the inability to remove stones with ERCP. One patient underwent T-tube placement, one patient underwent choledochoduodenostomy, and one patient underwent choledochojejunostomy after choledochotomy. In one patient, LCBDE was performed through the cystic duct due to suspected choledocholithiasis during the operation. One patient underwent laparoscopic primary repair for Strasberg type 4 bile duct injury, no bile leakage observed in the postoperative follow-up. One patient underwent cholecystectomy followed by Roux-en-Y hepaticojejunostomy for type 4 Mirizzi syndrome. Four patients underwent postoperative ERCP during index hospitalization: two of them had bile leakage, and two of them developed postoperative cholangitis related to choledocholithiasis. The postoperative complications included reoperation and ERCP for various reasons, such as bile leakage and recurrent choledocholithiasis. Factors affecting complications are presented in detail in Table 5.

### 3.6. Postoperative Follow-Up

In the postoperative period, 21 patients in the delayed cholecystectomy group and 4 in the early cholecystectomy group required ERCP. Of the 25 procedures, 13 were performed for the removal of a preoperatively placed biliary stent without postoperative complications. In 10 of the patients who needed ERCP at postoperative follow-up, biliary complaints due to choledocholithiasis were present. The majority of these complaints were jaundice due to recurrent common bile duct stones. Two patients required reoperation: one underwent choledochoduodenostomy for choledocholithiasis, and another had a second ERCP for cholangitis on postoperative day 6 and was reoperated on for septic shock following ERCP without detecting the intraoperative pathology.

## 4. Discussion

Choledocholithiasis can occur in up to 20% of patients with cholecystolithiasis [1]. Proper diagnosis and treatment of choledocholithiasis are crucial in preventing recurrent biliary events in this patient group, and selecting the appropriate treatment method is essential to reduce perioperative morbidity. However, there is currently no consensus on the optimal treatment of gallstones associated with choledocholithiasis.

Historically, common bile duct stones were treated with open choledochal exploration. However, with advancements in endoscopic techniques, ERCP has become the leading method for many years. ERCP can now be performed preoperatively, postoperatively, or intraoperatively using the rendezvous method, offering high treatment success. Sphincterotomy and stone extraction with ERCP have shown over 90% success in treating choledocholithiasis [18]. Additionally, laparoscopic common bile duct exploration (LCBDE) has replaced open choledochal exploration, reducing morbidity and demonstrating similar perioperative morbidity rates to preoperative ERCP while also lowering costs and shortening hospital stays [19,20]. However, postoperative bile leakage is more frequently observed in patients undergoing LCBDE [21]. Despite LCBDE and the laparoendoscopic rendezvous method (LERV) being recommended treatments, they are not widely adopted due to insufficient experience and difficulties in interdepartmental coordination. In our study, an open method was preferred in two cases due to the failure of preoperative ERCP.

Preoperative ERCP followed by laparoscopic cholecystectomy is currently the most widely used approach for treating choledocholithiasis. One common criticism of preoperative ERCP is the frequency of unnecessary procedures, reported to exceed 10% in the literature [22]. However, advances in diagnostic methods have reduced the use of ERCP for diagnostic purposes, limiting its application to patients with confirmed stones in the common bile duct via noninvasive tools such as magnetic resonance imaging. In our center, magnetic resonance cholangiopancreatography is routinely performed in patients with suspected choledocholithiasis, barring contraindications. Only 1.5% of the 197 patients in our study had no choledochal stones during ERCP.

Failure of ERCP in treating choledocholithiasis is another criticism, with failure rates reported at approximately 10% [23]. However, experienced gastroenterologists have significantly lower failure rates [24]. In our study, ERCP failed in only 3 of the 197 patients. Additionally, four patients were excluded due to failed cannulation. Another drawback of preoperative ERCP is the occurrence of recurrent biliary events (RBEs) during the waiting period between ERCP and cholecystectomy. Schiphorst et al. [9] reported RBEs in 20% of patients who were not operated on within 72 h after preoperative ERCP, with longer postoperative hospitalizations in patients who developed RBEs. Notably, 76% of the RBEs occurred at least one week after ERCP. Reinders et al. [11] found that RBEs occurred in 36.2% of patients who underwent cholecystectomy 6–8 weeks after ERCP, compared to only one patient in the early cholecystectomy group.

Early cholecystectomy can prevent RBEs and potential intraoperative complications by preoperatively detecting anatomical variations. Studies suggest that early cholecystectomy is superior to delayed cholecystectomy in preventing RBEs, reducing cholecystectomy-related morbidity, and lowering hospital costs [3,25]. Salman et al. [7] compared patients who underwent cholecystectomy within 24–72 h after ERCP to those operated on between 72 h and one week, finding shorter postoperative hospital stays and lower conversion rates in the early cholecystectomy group. A systematic review by Friis et al. [26], including 14 studies and 1930 patients, showed that cholecystectomy within the first 72 h reduced the conversion rates and could be safely performed within the first 24 h. In our study, we determined 72 h as the cut-off for early and late cholecystectomy. However, the mean time interval for the delayed cholecystectomy group is as long as 68.61 days. The reason for this difference is that early cholecystectomy is preferred to be performed in the first 3 days in our center, while cholecystectomy is generally preferred to be performed after 6–8 weeks in patients undergoing ERCP.

In our study, there was no significant difference in the postoperative length of stay, conversion rates, and postoperative biliary interventions between patients undergoing cholecystectomy within 72 h and those operated on after 72 h. Separate analyses for the first 24 and 48 h indicated that cholecystectomy can be safely performed early. The total length of stay increased with the prolongation of the interval between ERCP and LC, although this was not statistically significant.

Despite the literature supporting the safety and benefits of early cholecystectomy, delayed cholecystectomy is still frequently preferred in many institutions due to logistical and institutional practices. In their meta-analysis, Qi et al. [27] stated that the reason for this situation is that it requires the coordination of a multidisciplinary team with significant equipment, which is not practical for many healthcare systems. Mador et al. [28] found no effects of regional characteristics or socioeconomic status on the preference for early or late cholecystectomy, suggesting that the timing of LC after ERCP depends on institutional habits. In our study, 22.8% of patients underwent cholecystectomy within 72 h, while 49.75% preferred delayed cholecystectomy.

This study has limitations due to its retrospective nature, which makes it prone to type 2 errors. Since delayed cholecystectomy was generally preferred in our center, this led to a difference of 45 to 152 in the early and delayed cholecystectomy groups. Also, the heterogeneity of the patient group due to different indications for preoperative ERCP affects the results. Prospective, randomized controlled studies with larger and more homogeneous patient groups are needed to determine the optimal timing of cholecystectomy in patients treated with preoperative ERCP for choledocholithiasis.

## 5. Conclusions

This study suggested that early cholecystectomy is safe and feasible in patients undergoing ERCP for choledocholithiasis. Performing cholecystectomy within the first 24 h after ERCP does not increase perioperative morbidity and can help prevent recurrent biliary events. Further prospective studies are needed to confirm these findings and establish guidelines for the timing of cholecystectomy following ERCP.

## Figures and Tables

**Figure 1 healthcare-12-01407-f001:**
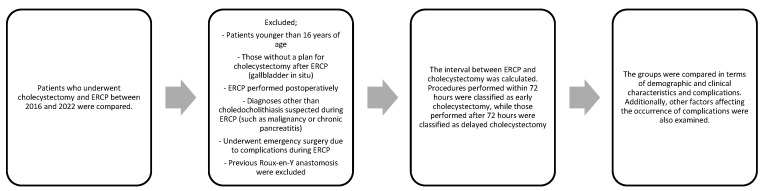
Flowchart of research strategy.

**Table 1 healthcare-12-01407-t001:** Preoperative characteristics of patients.

Sex	
Female	117 (59.4)
Male	80 (40.6)
Age (years)	56.46 ± 18.87
ASA score	
1	19 (9.6)
2	100 (50.8)
3	73 (37.5)
4	5 (2.5)
Biliary Presentation	
Acute cholecystitis	98 (49.7)
Acute cholangitis	41 (20.8)
Acute pancreatitis	46 (23.4)
Common bile duct diameter at MRCP (mm) Common bile duct diameter at ERCP (mm)	11.05 ± 9.2 11.55 ± 3.95
Preoperative ERCP sessions	
1	151 (76.6)
2	35 (17.8)
3	8 (4.1)
5	2 (1)
6	1 (0.5)
White blood cell count (×10^9^/L)	7.98 ± 2.81
C reactive protein (mg/L)	42.07 ± 56.51
Bilirubin (mg/dL)	1.21 ± 1.25
Alanine aminotransferase (U/L)	67.9 ± 146.89
Aspartate aminotransferase (U/L)	51.95 ± 118.16
Amylase (U/L)	109.94 ± 226.28

ASA: American Society of Anesthesiologists, MRCP: magnetic resonance cholangiopancreatography, ERCP: endoscopic retrograde cholangiopancreatography.

**Table 2 healthcare-12-01407-t002:** Early/delayed cholecystectomy patient profiles.

	Early Cholecystectomy	Delayed Cholecystectomy	*p* Value
Male/female	19:26	61:91	0.802
Age (years)	59.29	55.62	0.255
ASA score			0.269
1–2	24	95	
3–4	21	57	

ASA: American Society of Anesthesiologists.

**Table 3 healthcare-12-01407-t003:** Time interval and operative characteristics: ROC analysis.

	AUC (95% CI)	*p*
Operative complications	0.407 (0.256–0.558)	0.374
Operation related morbidity	0.543 (0.448–0.637)	0.395

**Table 4 healthcare-12-01407-t004:** Relationship between time interval and length of stay.

	Time from ERCP to Surgery (Days)	
	R *	*p*
Total LOS	−0.350	**<0.001**
PO LOS	−0.131	0.067

LOS: length of stay; PO: postoperative; *: Spearman’s rho correlation coefficient.

**Table 5 healthcare-12-01407-t005:** Factors affecting complications.

	Operative Complication		Univariate	
	Absent	Present	OR (95% CI)	*p*
LC first 24 h	35 (97.2)	1 (2.8)	0.629 (0.075–5.275)	0.669
LC first 48 h	41 (97.6)	1 (2.4)	0.516 (0.062–4.312)	0.541
LC first 72 h	44 (97.8)	1 (2.2)	0.471 (0.056–3.931)	0.487
Acute cholecystitis	92 (93.9)	6 (6.1)	3.163 (0.623–16.072)	0.165
Acute cholangitis	39 (95.1)	2 (4.9)	1.282 (0.249–6.6)	0.766
Acute pancreatitis	45 (97.8)	1 (2.2)	0.457 (0.055–3.816)	0.470
Sex				
Female	113 (96.6)	4 (3.4)		
Male	76 (95)	4 (5)	1.487 (0.361–6.127)	0.583
Biliary stent	41 (97.6)	1 (2.4)	0.516 (0.062–4.312)	0.541
Balloon dilatation	1 (100)	0 (0)	-	-
Age (years)	56.39 ± 18.58	58 ± 26.35	1.005 (0.967–1.044)	0.813
Common bile duct diameter at MRCP (mm)	10.93 ± 9.35	14 ± 3.37	1.019 (0.958–1.084)	0.542
Common bile duct diameter at ERCP (mm)	11.43 ± 3.93	14.38 ± 3.62	1.182 (1.003–1.392)	**0.046**
Preoperative ERCP sessions	1.33 ± 0.74	1.13 ± 0.35	0.473 (0.074–3.041)	0.431
White blood cell count (×10^9^/L)	7.96 ± 2.79	8.43 ± 3.4	1.056 (0.838–1.332)	0.643
C reactive protein (mg/L)	40.95 ± 55.11	56.91 ± 78.96	1.004 (0.99–1.019)	0.545
Creatinine (mg/dL)	0.82 ± 0.41	0.84 ± 0.26	1.097 (0.224–5.37)	0.909
Bilirubin (mg/dL)	1.23 ± 1.27	0.73 ± 0.37	0.348 (0.062–1.955)	0.231
Alanine aminotransferase (U/L)	69.1 ± 149.8	39.63 ± 22.01	0.996 (0.984–1.009)	0.567
Aspartate aminotransferase (U/L)	52.55 ± 120.56	37.75 ± 18.65	0.998 (0.984–1.012)	0.732
Alkaline phosphatase (U/L)	144.29 ± 121.77	153.38 ± 159.44	1.001 (0.995–1.006)	0.838
Amylase (U/L)	112.29 ± 232.98	72.33 ± 34.82	0.998 (0.985–1.011)	0.707
Biliary hospitalization	2.8 ± 2.1	3.25 ± 1.04	1.079 (0.838–1.389)	0.555
Time interval (days)	54.25 ± 65.36	26.38 ± 27.44	0.988 (0.968–1.008)	0.234

LC: laparoscopic cholecystectomy, MRCP: magnetic resonance cholangiopancreatography, and ERCP: endoscopic retrograde cholangiopancreatography.

## Data Availability

The data supporting the findings of this study are available in the following data set: Büyükkasap, Ç. (2024). Early cholecystectomy after endoscopic retrograde cholangiopancreatography is feasible and safe [Data set]. Zenodo. https://doi.org/10.5281/zenodo.11214236, accessed on 27 June 2024. The data can be accessed via the provided link.

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
