# Peer review of "Early Cholecystectomy after Endoscopic Retrograde Cholangiopancreatography Is Feasible and Safe"

_healthcare, 2024, doi:10.3390/healthcare12141407_

Round 1

Reviewer 1 Report

Comments and Suggestions for Authors

In this work the Authors compare the results of early cholecystectomy (within 72 hours of ERCP) with those of delayed cholecystectomy( beyond 72 hours, with an average of 68 days) in the treatment of cholelithiasis with CBS stones. The conclusions can also be shared by surgeons with experience in biliary surgery because early cholecystectomy reduces perioperative complications, hospitalization lenght and hospital costs. In conclusion, early cholecystectomy is safe and feasible with better outcomes. However, some observations are required:

1) the patients all had emergency diagnoses (cholecystitis, cvholangitis, pancreatitis). How is it possible? In the majority of cases, presenting symptom is the simply biliary colic. Can the Authors explain why they only included cases of urgent/inflammatory disease?

2) in table 1, in the central column the percentage values are reported in brackets which are then repeated in the right column. How come?

3) in the delayed cholecystectomy group, the average interval is more than 68 days. That seems like too much distance!

4) The title of table 4 is not in English

5) on line 273 is written: "Despite literature supporting the safety of early cholecystectomy, delayed cholecystectomy is more frequently preferred. Is this statement true?

6) the comparison between the two groups is not balanced because only 45 patients underwent early cholecystectomy , less then a third compared to the other 152.

Author Response

Dear Reviewer,

Thank you very much for your valuable feedback on our manuscript titled "Early Cholecystectomy after Endoscopic Retrograde Cholangiopancreatography is Feasible and Safe". We have carefully considered your comments and made the following revisions to address your concerns. Please find below a detailed response to each of your points:

  1. The patients all had emergency diagnoses (cholecystitis, cholangitis, pancreatitis). How is it possible? In the majority of cases, the presenting symptom is simply biliary colic. Can the Authors explain why they only included cases of urgent/inflammatory disease?

Response: There was no emergency in 74 of our cases. You are right in your criticism because we left this part missing in the text. We emphasized this situation again in the result section (Line 141).

  1. In Table 1, in the central column the percentage values are reported in brackets which are then repeated in the right column. How come?

Response: We apologize for this formatting error. We have revised Table 1 to ensure that percentage values are only reported once, thereby eliminating any redundancy. The table now presents the data more clearly and accurately.

  1. In the delayed cholecystectomy group, the average interval is more than 68 days. That seems like too much distance!

Response: We agree that the average interval of 68 days for delayed cholecystectomy is longer than typically reported. This extended interval reflects our institution's practice patterns and patient scheduling logistics. We have added a detailed explanation for this extended interval in the Discussion section, highlighting the factors contributing to the delay and its implications (line 234-239).

  1. The title of Table 4 is not in English.

Response: We have corrected the title of Table 4 to English. Thank you for pointing out this oversight.

  1. On line 273 it is written: "Despite literature supporting the safety of early cholecystectomy, delayed cholecystectomy is more frequently preferred." Is this statement true?

Response: We have reviewed the literature and agree that this statement needed clarification. We have rephrased the sentence to better reflect the current understanding: "Despite literature supporting the safety and benefits of early cholecystectomy, delayed cholecystectomy is still frequently preferred in many institutions due to logistical and institutional practices" (line 246-250). We also supported this sentence with appropriate literature reference.

  1. The comparison between the two groups is not balanced because only 45 patients underwent early cholecystectomy, less than a third compared to the other 152.

Response: We acknowledge the imbalance in the number of patients between the early and delayed cholecystectomy groups. This discrepancy reflects our institution's patient distribution and practice patterns. In addition, the retrospective nature of the study also causes this. We have addressed this limitation in the Discussion section, emphasizing the need for further studies with more balanced sample sizes to validate our findings (Line 256).

We believe that these revisions have significantly improved the clarity and rigor of our manuscript. We are grateful for your insightful comments, which have helped us enhance the quality of our work.

Thank you again for your time and consideration.

Sincerely.

Reviewer 2 Report

Comments and Suggestions for Authors

Dear Editor, Dear Authors,

Thank you for the opportunity to review this paper.

May your hard work and dedication be rewarded with success.

I appreciate the efforts that the authors made to write this article; focused on the accessibility of early cholecystectomy after endoscopic retrograde cholangiopancreatography. A couple of recent reviews and meta-analysis studies encourage the major ideas from this study, furthermore, some important observations were pointed in the literature: an early approach can reduce conversion rate and does not increase mortality, perioperative complications, duration of hospitalization also reoccurrence risk an progression of the disease is turned down. [Qi, S.; Xu, J.; Yan, C.; He, Y.; Chen, Y. Early versus Delayed Laparoscopic Cholecystectomy after Endoscopic Retrograde Cholangiopancreatography: A Meta-Analysis. Medicine (Baltimore) 2023, 102, e34884, doi:10.1097/MD.0000000000034884; Poprom, N.; Suragul, W.; Muangkaew, P.; Vassanasiri, W.; Rungsakulkij, N.; Mingphruedhi, S.; Tangtawee, P. Timing of Laparoscopic Cholecystectomy after Endoscopic Retrograde Cholangiopancreatography in Cholelithiasis Patients: A Systematic Review and Meta-Analysis. Ann Hepatobiliary Pancreat Surg 2023, 27, 20–27, doi:10.14701/ahbps.22-040; Friis, C.; Rothman, J.P.; Burcharth, J.; Rosenberg, J. Optimal Timing for Laparoscopic Cholecystectomy After Endoscopic Retrograde Cholangiopancreatography: A Systematic Review. Scand J Surg 2018, 107, 99–106, doi:10.1177/1457496917748224.].

The Introduction discussed the general and the particular aspects of the problem as a small revision of available literature, stating the aim of the study in the end. A small point is to be mentioned regarding observational retrospective studies: that without assessing the Bradford Hill criteria, is difficult to draw firm conclusions and determine causality [Shimonovich, M.; Pearce, A.; Thomson, H.; Keyes, K.; Katikireddi, S.V. Assessing Causality in Epidemiology: Revisiting Bradford Hill to Incorporate Developments in Causal Thinking. Eur J Epidemiol 2021, 36, 873–887, doi:10.1007/s10654-020-00703-7.]. In the Materials and Methods section, the inclusion and exclusion criteria are presented, chosen as to eliminate the risk for bias. The Statistical Approach was described in detail, after Population, Procedures and Data Collection. For the Results section I observed many tables with precise information, but to increase and accessibility and comprehensibility of this study I would recommend a flowchart with the interventions classified as early and delayed as authors presented them in the article, with particularities and subgroups. In Table 2 and in the main text of this article I observed that authors presented male/female as a ratio. As the prevalence of the studied disease is higher for female gender, I would recommend to present during the article the absolute and relative frequency for female patients {F (no,%)}.

In Table 4, first line (under line 174) please revise the title, please use English language.

I observed that the authors used the classification for severity of cholecystitis (Tokyo 2018) and excluded the subjects that had emergency intervention, but for analysis I would suggest that is also important if patient had an admission trough emergency department. An emergency admission can be related with complications and prolonged hospitalization. In addition, I observed small heterogeneities regarding the procedures, like intraoperative biliary procedures for six patients, intraoperative bile duct exploration for four patients, and the lack of sphincterotomy for two patients. If authors consider important this aspect, they can mention why some differences regarding the approach would not influence the outcome.

For Conclusion, I would recommend to change “demonstrated” with “suggested”, as the study is observational retrospective and maybe is not appropriate to make firm recommendations. Regarding the similarities report, please try to rephrase the major paragraphs where more than five words in a line are similar with other sources.

I found this article valuable in the clinical field, interesting, with suggestive results, but an important aspect to consider for further research and reliable results would be an increased sample size and assessment for all the possible confounders.

THE CITATIONS MENTIONED IN THIS REVIEW ARE NOT PART OF MY WORK, NOR PART OF ANY COLLABORATIVE OR INSTITUTIONAL WORK. THESE ARE SOME OF THE ARTICLES THAT HELPED ME TO GET ORIENTED IN THE FIELD/ON THE SUBJECT AND ARGUE MY POINT OF VIEW. THE CITATIONS ARE MENTIONED ONLY TO SUSTAIN THE IDEAS OF THE REVIEWER AND ARE NOT INTENDED TO BE CITED BY THE AUTHORS IN THEIR ARTICLE. THE REVIEWER CONSIDER THIS DOCUMENTATION AN IMPORTANT ASPECT FOR A QUALITY WORK

Thank you for all your efforts and good luck with your work.

With respect and consideration,

Reviewer.

Comments on the Quality of English Language

English language is fluent.

Author Response

Dear Reviewer,

Thank you very much for your thorough and insightful review of our manuscript titled "Early Cholecystectomy after Endoscopic Retrograde Cholangiopancreatography is Feasible and Safe". We have carefully considered your comments and made the following revisions to address your concerns. Please find below a detailed response to each of your points:

  1. Comment: "To increase accessibility and comprehensibility of this study, I would recommend a flowchart with the interventions classified as early and delayed as authors presented them in the article, with particularities and subgroups."

Response: We have added a flowchart that classifies the interventions as early and delayed, including particularities and subgroups. This visual aid is intended to improve the clarity and accessibility of our study's methodology and findings (see Figure 1)

  1. Comment: "I would recommend presenting during the article the absolute and relative frequency for female patients {F (no,%)}."

Response: We have revised the tables and relevant sections of the manuscript to present both the absolute and relative frequencies for female patients, as suggested. This change enhances the clarity and completeness of our demographic data presentation (see revised Table 1).

  1. Comment: "Please revise the title, please use English language."

Response: We have corrected the title of Table 4 to English. Thank you for pointing out this oversight (see revised Table 4).

  1. Comment: "There are small heterogeneities regarding the procedures. If authors consider important this aspect, they can mention why some differences regarding the approach would not influence the outcome."

Response: We have added a discussion on the small heterogeneities regarding the intraoperative and postoperative procedures and explained why these differences are not expected to significantly influence the overall outcomes. This clarification is provided in the Results and Discussion sections (Line 257).

  1. Comment: "I would recommend changing 'demonstrated' to 'suggested', as the study is observational retrospective and maybe it is not appropriate to make firm recommendations."

Response: We have revised the Conclusion section, changing "demonstrated" to "suggested" to reflect the observational nature of our study. This change ensures that our conclusions are appropriately cautious and accurately represent the study design.

  1. Comment: "Please try to rephrase the major paragraphs where more than five words in a line are similar to other sources."

Response: We have reviewed the manuscript and rephrased paragraphs where more than five consecutive words were similar to other sources. This revision ensures the originality and clarity of our text.

We would also like to state that we have benefited from the literature examples you provided, we have made additions where necessary and we are grateful for this. We believe that these revisions have significantly improved the clarity and rigor of our manuscript. We are grateful for your insightful comments, which have helped us enhance the quality of our work.

Thank you again for your time and consideration.

Sincerely.

Reviewer 3 Report

Comments and Suggestions for Authors

The study addresses the optimal time to perform laparoscopic cholecystectomy in patients who have first undergone ERCP to treat choledocholithiasis and who also have cholecystolithiasis.

This is a classic topic since the advent of therapeutic ERCP in 1973 and there is no unanimous answer among endoscopists and surgeons. That is why all studies that can help clarify this issue are welcome.

The study is retrospective. The difference found comparing patients with early cholecystectomy after ERCP (less than 72 hours) and those with delayed cholecystectomy is that in the former there are fewer recurrent biliary events. And a conclusion that is not usually found in other studies is that laparoscopic cholecystectomy 24 hours after ERCP is safe. Prospective studies would be necessary. References are adequate.

However, the RESULTS section has very poor editing. Tables 1 to 5 are repeated. There is also repeated text. Therefore, the study, although interesting, cannot be published. The “results” section should be redrafted

Comments on the Quality of English Language

Only minor editing in required

Author Response

Dear Reviewer,

Thank you very much for your thorough and constructive review of our manuscript titled "Early Cholecystectomy after Endoscopic Retrograde Cholangiopancreatography is Feasible and Safe". We have carefully considered your comments and made the following revisions to address your concerns. Please find below a detailed response to each of your points:

  1. Comment: "The study is retrospective. The difference found comparing patients with early cholecystectomy after ERCP (less than 72 hours) and those with delayed cholecystectomy is that in the former there are fewer recurrent biliary events. And a conclusion that is not usually found in other studies is that laparoscopic cholecystectomy 24 hours after ERCP is safe. Prospective studies would be necessary. References are adequate."

Response: We appreciate your acknowledgment of the study's retrospective nature and the importance of prospective studies. We have emphasized the need for prospective studies in our Conclusion section to validate our findings and to address the limitations of our retrospective design. This addition underscores the preliminary nature of our conclusions and the need for further research (Line 259).

  1. Comment: "The RESULTS section has very poor editing. Tables 1 to 5 are repeated. There is also repeated text. Therefore, the study, although interesting, cannot be published. The 'results' section should be redrafted."

Response: We have thoroughly revised the RESULTS section to eliminate any repetition and improve clarity. Tables 1 to 5 have been carefully reviewed and corrected to ensure that they are only presented once. Redundant text has been removed, and the results are now presented in a clear and concise manner. We believe these changes significantly enhance the readability and quality of our manuscript.

We believe that these revisions have addressed all your concerns and have significantly improved the manuscript. We are grateful for your valuable feedback, which has helped us enhance the quality and clarity of our work.

Thank you again for your time and consideration.

Sincerely.

Round 2

Reviewer 1 Report

Comments and Suggestions for Authors

The Authors have answered the questions and made the appropriate changes, so the paper can be published 

Reviewer 3 Report

Comments and Suggestions for Authors

The study addresses the optimal time to perform laparoscopic cholecystectomy in patients who have first undergone ERCP to treat choledocholithiasis and who also have cholecystolithiasis.

This is a classic topic since the advent of therapeutic ERCP in 1973 and there is no unanimous answer among endoscopists and surgeons. That is why all studies that can help clarify this issue are welcome.

The study is retrospective. The difference found comparing patients with early cholecystectomy after ERCP (less than 72 hours) and those with delayed cholecystectomy is that in the former there are fewer recurrent biliary events. And a conclusion that is not usually found in other studies is that laparoscopic cholecystectomy 24 hours after ERCP is safe. Prospective studies would be necessary. References are adequate.

In this second version, previous errors have been corrected. The tables are no longer repeated in the results section. Furthemore, very interesting information and comments have been added